# Inner Surface Hydrophilic Modification of PVDF Membrane with Tea Polyphenols/Silica Composite Coating

**DOI:** 10.3390/polym13234186

**Published:** 2021-11-30

**Authors:** Qiang Xu, Xiaoli Ji, Jiaying Tian, Xiaogang Jin, Lili Wu

**Affiliations:** 1School of Materials Science and Engineering, Wuhan University of Technology, Wuhan 430070, China; qiangxu@whut.edu.cn (Q.X.); jxl@whut.edu.cn (X.J.); lovertjy@163.com (J.T.); 2Advanced Engineering Technology Research Institute of Zhongshan City, Wuhan University of Technology, Xiangxing Road 6, Zhongshan 528400, China

**Keywords:** polyvinylidene fluoride, surface modification, tea polyphenols, superhydrophilic

## Abstract

The use of Polyvinylidene fluoride (PVDF) membranes is constrained in wastewater treatment because of their hydrophobic nature. Therefore, a large number of researchers have been working on the hydrophilic modification of their surfaces. In this work, a superhydrophilic tea polyphenols/silica composite coating was developed by a one-step process. The composite coating can achieve not only superhydrophilic modification of the surface, but also the inner surface of the porous PVDF membrane, which endows the modified membrane with excellent water permeability. The modified membrane possesses ultrahigh water flux (15,353 L·m^−2^·h^−1^). Besides this, the modified membrane can realize a highly efficient separation of oil/water emulsions (above 96%).

## 1. Introduction

With the rapid development of modern industry, we will face more and more challenges in environmental and energy crises, among which the most important is the pollution of water resources. Nowadays, large amounts of wastewater are generated in industrial production of goods such as food, textiles, petroleum, petrochemical and metal processing [1]. The researchers used flotation [2], coagulation [3], biological treatment [4], membrane separation [5] and other methods for treatment. Among these methods, membrane separation technology is considered to be the most effective method to solve the wastewater problem [6,7,8,9]. Membrane separation technology has very high efficiency in treating sewage, and has a certain degree of selectivity and stability. So far, there are many kinds of membranes that can be used to treat wastewater, such as polymer membranes [10], mixed matrix membranes [11], ceramic membranes [12] and reticulated polymer membranes [13]. Polymer membranes, as a common material, have simple manufacturing conditions and are suitable for large-scale industrial production compared with other types of membranes. They are selected for the treatment of oily wastewater [14]. Nevertheless, the lifespan of the membrane is short due to being typically hydrophobic, which may lead to severe fouling by organic matter. Therefore, hydrophilic modification of the hydrophobic membrane is of vital importance. Many efforts such as blending [15,16,17,18], grafting [19,20], co-polymerization [21,22,23,24] and layer-by-layer assembly [25,26,27] methods have been devoted to improving the hydrophilicity of the membrane. However, these methods may require special equipment or complicated conditions, and have certain limitations in practical applications.

Surface coatings have received ever-increasing attention in the field of membrane hydrophilic modification due to their simplicity and being environmentally friendly [28,29,30]. Containing abundant catechol/pyrogallol moieties, tea polyphenol is a suitable material for surface coating, since it possesses outstanding surface binding affinity and benign hydrophilicity [31,32]. Green tea discarded after daily use still contains tea polyphenols. The remaining tea polyphenols can be extracted from the green tea residues, which not only make the best use of the material for actual production, but also save costs for enterprises. This method has certain economic benefits. Similar to the heuristic surface functionalization of mussels containing catechol polymers, almost all types of substrates can be surface coated with tea polyphenol molecules regardless of shape and size [33]. Sileika et al. [34] reported this landmark study. They successfully coated tea polyphenols and analogs such as epigallocatechin-3-gallate (EGCG), pyrogallol (PG), and tannic acid (TA) on various organic and inorganic substrates to functionalize it. The substrate usually exhibits antibacterial and antioxidant properties. Zhang et al. [33] modified polyethersulfone (PES) ultrafiltration membranes by co-deposition of EGCG and polyethyleneimine (PEI). The results show that the 600-molecular-weight modified membrane has 99% rejection of Congo Red, and exhibits good resistance to organic solvents and structural stability. At the same time, it has proved its great potential in the treatment of textile waste liquid.

There are numerous studies of using tea polyphenols to modify the membrane surface [33]. These modification methods only make the membrane surface more hydrophilic, and do not penetrate deeply into the membrane pores to achieve hydrophilic modification of the inner surface of the membrane. The surface topology of SiO_2_ is a 3D chain, which contains two forms of siloxane (Si-O-Si) and silanol (Si-OH) [34]. These two groups give the surface of SiO_2_ excellent hydrophilicity. Therefore, nano-SiO_2_ is often used in the surface modification of membranes. Thus, nano-silica was chosen as a secondary functional material to further improve the properties of the membrane. Due to the small size and fast deposition rate of nano-silica, low levels of tea polyphenols can be brought into the pores of the membrane to complete the hydrophilic modification of the inner surface of the membrane, which allows water to flow through the membrane easily and quickly. With the hydrophilic composite coating depositing on both the surface and inside of the membrane, the hydrophilicity of the modified membrane will be greatly enhanced.

In this work, a facile strategy based on tea polyphenols and nano-silica composite coating was developed to achieve the hydrophilic transformation of hydrophobic PVDF membranes. The coating schematic diagram is shown in Figure 1. The composite coating can enter the membrane pores to complete the hydrophilic modification of the inner surface, which endows the modified membrane with ultrahigh water flux and excellent emulsion separation performance. We believe that this modification strategy is a good solution for constructing advanced filtration materials for practical oil/water separation and that it helps to improve the living environment.

## 2. Experimental

First, green tea (2 g) mixed with deionized water (120 mL, pH = 3, adjusted by hydrochloric acid) was shocked (130 r/min) in the oscillator at 60 °C for 50 min. Then, the solution was filtered to obtain the green tea solution. Next, the pH of the green tea solution was adjusted to 7.8 by adding tris(hydroxymethyl)aminomethane, and a certain amount of silica solution (0.015 g, 0.03 g and 0.06 g) was dropped into it, stirring at a low speed for 30 min to mix well. In the end, the pre-treated PVDF membranes (immersed in isopropyl alcohol for 30 min to clean the surface and washed with deionized water to exchange the isopropyl) were immersed in the modified solution for 6 h in the atmosphere at 25 °C. The pre-treated membrane that was immersed in deionized water for 6 h could be named M0, the pre-treated membrane that was only modified by tea polyphenols for 6 h could be named as M1, and the pre-treated membranes modified by adding 0.015 g, 0.03 g and 0.06 g silica solution to the green tea solution could be named as M2, M3 and M4 respectively. The chemical composition of the modified membrane was characterized by the contact angle test system (JC2000C, Zhongchen, China), X-ray photoelectron spectroscopy (XPS, ESCALAB250Xi, Britain) and a field emission scanning electron microscope (FESEM, Zeiss, Ultra Plus, Germany).

## 3. Results and Discussion

### 3.1. Analysis of XPS Spectrum and Morphology

XPS was used to identify the existence of TP/SiO_2_ coating on the modified membrane and for the characterization of the chemical composition of the modified membrane. From Appendix A, a large number of F 1s was shown in M0. However, the number of F 1s decreased while the peak intensity of O 1s and C 1s increased dramatically for M1 and M3, indicating that the tea polyphenols coating had been deposited on the membrane. In addition, M3 had an extra peak at 100.8 eV and 153.8 eV, which represent Si 2p and Si 2s, respectively, indicating that SiO_2_ had been successfully coated on the surface of the membrane. It can be seen from Table 1 that the surface of the unmodified film M0 contains a large amount of F. After the modification of tea polyphenol alone, the element F on M1 is greatly reduced. But the F element on M3 is less than that on M1, indicating that the modified coating on M3 is thicker and denser. It is worth noting that the O element on M3 was about 4% greater than that on M1. On the one hand, SiO_2_ was successfully coated on the surface of the membrane, resulting in an increase in the O element; on the other hand, the introduction of SiO_2_ may have caused more tea polyphenol to be deposited on the membrane, which increased the levels of the O element.

The morphology of the modified membrane cross section is shown in Figure 2. From the first column, the membrane matrix of M0 was uniform and the overall morphology remained the same. However, the membrane matrix in the upper part of M1 was not uniform, while M3 had a large section of very obvious modification traces, which was staggered and uneven. From the second column, the modification depth of M1 was only 8.33 μm, while that of M3 reached 15.18 μm. Finally, from the third column, the M0 membrane showed the typical sharp polymer structure. The membrane matrix of M1 obviously had a layer of coating on it, but it was very smooth and delicate. The membrane matrix of M3 was stacked up layer by layer like tree bark, which could greatly increase the roughness of inner surfaces and formed some micro-nano structures helping water molecules to pass through quickly. In addition, it can be seen from Appendix A that the porosity of the modified membrane is reduced compared with M0. At the same time, compared with the M2 and M3 membranes, the porosity on the M4 membrane is greatly reduced. Furthermore, it can even be seen that the membrane pores are completely blocked, which will cause the water flux of the M4 membrane to decrease. This is consistent with the subsequent changes in water flux. The EDX images of the cross section of M3 was shown in Appendix A. O and Si elements were evenly distributed on the cross section of the membrane, indicating that SiO_2_ could indeed carry some TP into the membrane pores and modify the inner surfaces.

### 3.2. Hydrophilicity and Separation Performance of Modified PVDF Membrane

Due to the hydrophilicity of TP and SiO_2_, it can turn hydrophobic porous channel surfaces to hydrophilic surfaces, which allow water to pass through the porous channel quickly (Figure 3a). This can also be demonstrated in Figure 3b. The water contact angle of M0 was 109°. The water contact angle of M1 dropped to 52°, but they are still a long way from being superhydrophilic. Interestingly, the water contact angles of M2, M3 and M4 reached 30°, 24° and 33°, respectively, exhibiting superhydrophilic properties. Besides this, Figure 3e showed the water permeation rates of each modified membrane; the water contact angle of M0 changed to 92° after 11 s, showing an inherent hydrophobic nature. The water contact angle of M1 changed to 0° after 3 s, indicating its good hydrophilicity, while the water contact angles of M2, M3 and M4 all changed to 0° after 1 s. This indicated that the modified substance had entered the internal matrix of the membrane to greatly increase the water penetration rate. However, Zhang et al. modified the PVDF membrane with tannin acid. In this situation, it takes 4 s for the water contact angle to decrease from 33° to 0°. [35] Liu et al. used tannin acid and polyethyleneimine to modify the PVDF membrane, and the water contact angle was around 40°. [36] The comparison also shows the super-hydrophilicity of the co-deposited modified film in this experiment. Figure 3b showed the pure water flux of each modified membrane; the pure water flux of the M3 membrane reached 15,353 L × m^−2^ × h^−1^, which was more than twice that of M0 (7507 L × m^−2^ × h^−1^). This indicated that the hydrophilicity of the surface and the inside had been greatly improved.

The modified membrane can also effectively handle oil/water emulsion. Figure 3c shows the UV absorption spectra of the filtrate solution. At 261 nm was the UV absorption peak of toluene. It could be seen from the figure that the absorbance of M0 was around 1.1, while the absorbance of M1 was around 0.7. The absorbance of the membrane modified by TP/SiO_2_ was lower than that of both M0 and M1 membranes, indicating that their filtrate contained less toluene. It is worth noting that the M3 membrane had the lowest absorbance, which indicated that it had the best performance in retaining emulsions, and this result was consistent with the conclusion of hydrophilicity above. Because the M3 membrane displayed the best hydrophilicity, water molecules could easily form a complete hydration layer on the membrane surface to prevent oil droplets from passing through. Based on the absorbance, the retention rate of each modified membrane could be calculated, as shown in Figure 3d.

The toluene retention rate of M0 was 94% and the retention rate of M1 was slightly improved, while the rate of the TP/SiO_2_ modified membrane was around 97%. As can be seen from the electronic photo on the top left, the pure white emulsion became clear and transparent after treatment of the membrane.

### 3.3. Antifouling and Stability Performance of Modified PVDF Membrane

The work performed by separating the combined two phases into two independent phases is called adhesion work (*W_SL_*). The adhesion work between oil droplets and the film surface can be calculated from the Young-Dupre formula [37]:(1)$$W_{SL}=\sigma_{L}\;\left(1+\cos\theta \right)\;$$

$\sigma_{L}$ represents the surface tension of the liquid, and *θ* represents the contact angle.

This formula combines the contact angle with the adhesion work, and the range of the contact angle is 0° to 180°. The larger the contact angle, the smaller the adhesion work and the better the adhesion resistance of the solid surface. It can be seen from Table 1 that the adhesion work of M0 is 18.07 mN/m. After coating with tea polyphenols, the adhesion work of M1 dropped to 3.12, which was nearly six times lower than that of M0. After the addition of the SiO_2_ co-deposition modification, the adhesion work of M2, M3, M4 further decreased, indicating that the co-deposition modified film of tea polyphenols and silica sol has better anti-adhesion properties. The adhesion work of M3 was only 1.70 mN/m. The antifouling performance of the surface was further tested by rinsing after soaking in soybean oil. [38] As can be seen in Appendix A, compared to M0 and M1, M3 did not have red soybean oil adhering to the surface when immersed in water. After the M3 membrane was vertically removed from the water, the soybean oil on the surface was easily rinsed off by pure water, whereas the M0 and M1 membranes would have residual red soybean oil on the surface that could not be rinsed off, indicating that the M3 membrane surface has excellent oil adhesion resistance and self-cleaning performance.

From Appendix A, it can be seen that the film can still maintain a stable water flux and underwater oil contact angle in an acidic environment. The stability of the M3 membrane in pure water, 10 g/L NaCl, and 10 g/L Na_2_SO_4_ solutions is shown in Figure 4. It can be seen from Figure 4a that after one day of immersion in pure water, 10 g/L NaCl or 10 g/L Na_2_SO_4_ solution, the underwater oil contact angle of the M3 membrane remained above 150°, indicating the M3 membrane still has underwater super-oleophobicity. In addition, the rejection rate of the toluene emulsion of the M3 membrane is also maintained above 96%, which proves that the M3 membrane can effectively treat the toluene emulsion. Finally, it can be seen from Appendix A that the crystallinity and mechanical properties of the film did not change significantly before and after the modification.

From Figure 4b–d, it can be seen that, after a day of washing with pure water, the oil droplets will deform during the ascent process. It shows that there will be a small amount of coating peeling off of the M3 film in pure water, causing its surface to adhere to oil droplets. After washing with 10 g/L NaCl and 10 g/L Na_2_SO_4_ solutions for one day, the oil droplets remained spherical in the process of descending contact and rising from the surface of the membrane. The above shows that the tea polyphenol/silica coating is very stable and has excellent separation performance and anti-fouling performance.

## 4. Conclusions

We used tea polyphenols and silica to achieve hydrophilic modification of the inner surface of the PVDF membrane. By studying the surface and cross-sectional morphology of the modified PVDF membrane, it was found that silica can be deposited into the pores of the PVDF membrane together with the tea polyphenol polymer. It realizes the modification of the internal matrix of the membrane pores and constructs a unique hydrophilic channel. In addition, the Si element is evenly distributed on the surface and cross section of the modified membrane, indicating that the tea polyphenol/silica composite coating can be evenly deposited on the membrane surface, which greatly improves the hydrophilicity of the membrane. Furthermore, oil-water emulsion can be handled well. Due to the introduction of nano-SiO_2_, the surface oil resistance of the modified film has been enhanced. Its underwater oil contact angle even reaches 160°, and the retention rate of toluene emulsion is about 96%. In addition, the surface of the modified film has very low adhesion to soybean oil, and the remaining soybean oil on the surface can also be easily washed with pure water, indicating that the surface of the modified film has excellent oil adhesion resistance and self-cleaning properties.

## Figures and Tables

**Figure 1 polymers-13-04186-f001:**
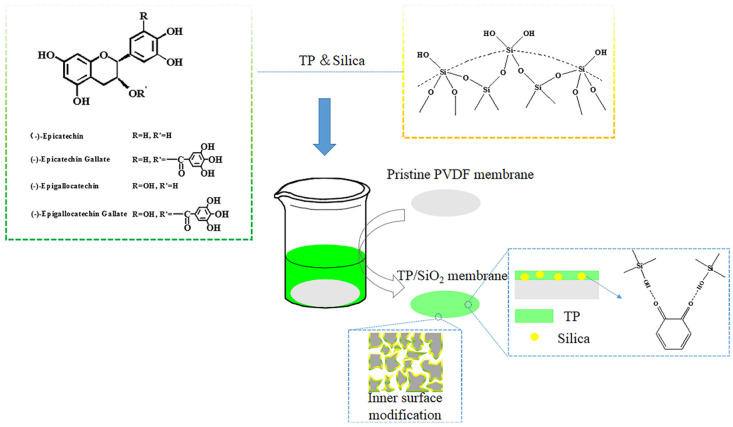
The coating schematic diagram of TP/SiO_2._

**Figure 2 polymers-13-04186-f002:**
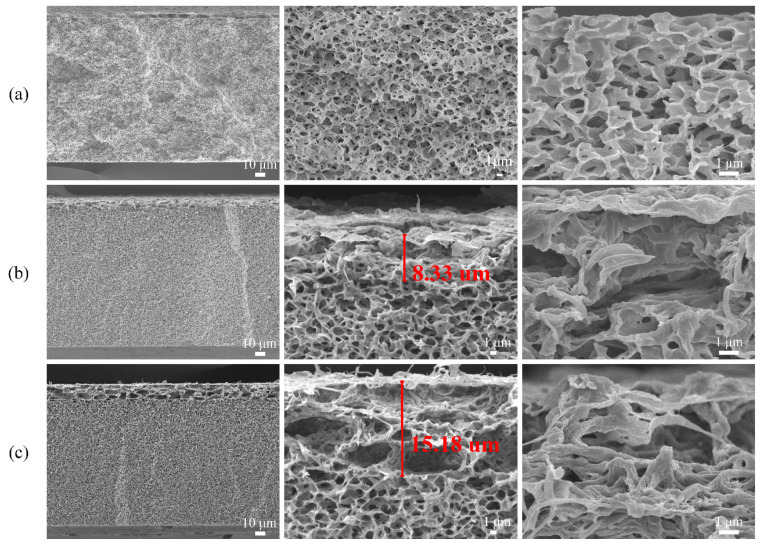
SEM images of the cross section of modified membrane: (**a**) M0, (**b**) M1, (**c**) M3.

**Figure 3 polymers-13-04186-f003:**
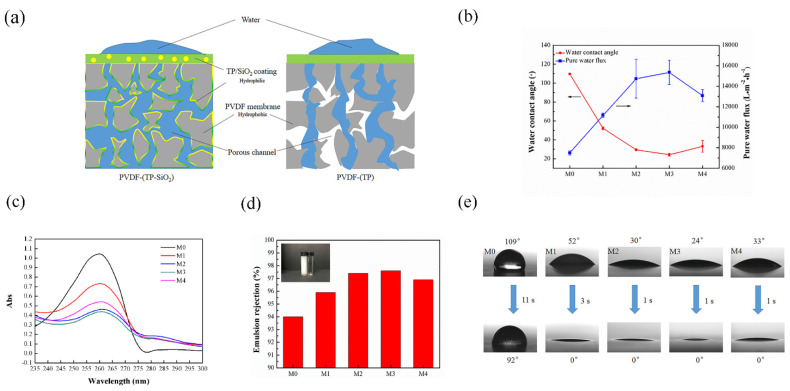
(**a**) The mechanism of water passing through the modified membrane, (**b**) Water contact angle and pure water flux of modified membrane, (**c**) UV-Vis spectra of the toluene emulsion filtrate of modified membrane, (**d**) Emulsion rejection of modified membrane, (**e**) Images of a water drop on different membranes.

**Figure 4 polymers-13-04186-f004:**
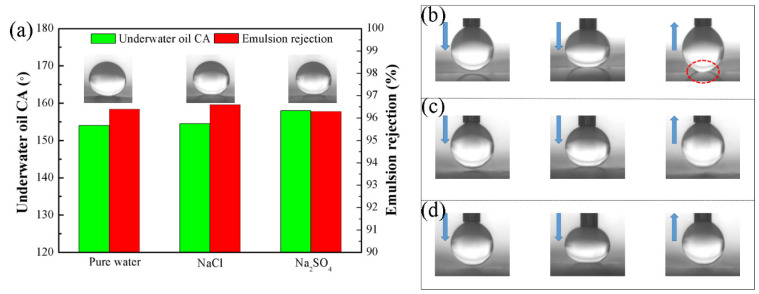
(**a**) Underwater oil CA and emulsion rejection rate of M3 membrane after washing in different solutions, (**b**) Dynamic underwater oil adhesion test of M3 membrane after washing in pure water, (**c**) Dynamic underwater oil adhesion test of M3 membrane after washing in NaCl solution, (**d**) Dynamic underwater oil adhesion test of M3 membrane after washing in Na_2_SO_4_ solution.

**Table 1 polymers-13-04186-t001:** The adhesion work of film M0, M1, M2, M3, M4.

	UOCA/(°)	Adhesion Work/(mN/m)
M0	111	18.07
M1	153	3.12
M2	158	2.05
M3	160	1.7
M4	159	1.87

## Data Availability

Not applicable.

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
