# Peer review of "Inner Surface Hydrophilic Modification of PVDF Membrane with Tea Polyphenols/Silica Composite Coating"

_polymers, 2021, doi:10.3390/polym13234186_

Round 1
Reviewer 1 Report
The manuscript "Inner surface hydrophilic modification of PVDF membrane with tea polyphenols/silica composite coating" shows an interesting concept making PVdF hydrophilic by coating and soaking tea phenol with SiO2. The concept is nice unfortunately the science behind such is a little missing.
- The authors need to add a proper characterization such as FTIR showing they obtained the composites. Please add those and that can be shown in supplementary. Additionally if you have larger pores as seen in SEM the modulus of such membrane will change please present strain stress results
- If you show results and discussion there should be a discussion to other work make in the field as example contact angles. Please provide such hence the paper missing scientific depth
- The authors talk a lot about hydrophilic behavior inside PVdF membrane but no analysis made to confirm such.
- There a lot of typos in the manuscripts with "." missing some letter writing big and grammatical errors as well. Please let the manuscript check with a native speaker
- If you show this as communication Figure 4 is somehow not real scientific please add some data for such and show Figure 4 in supplementary.
- There a lot of polymers as chitosan which are hydrophilic in nature why you need to use PVdF. Please give reason for such as well in motivation of work.
- The conclusion need be rewritten its far to shallow and you should present your major results
Reviewer 2 Report
In this work, a superhydrophilic tea polyphenols/silica com-
posite coating was developed by a one-step process. The authors reported composite coating can achieve not only superhydrophilic modification of the surface, but also the inner surface of the porous PVDF membrane, which endows the modified membrane with excellent water permeability.
The whole work is very interesting.
POINTS FOR IMPROVEMENT:
- The chemical stability of the prepared membranes was not examined. For example pH, presence of an oxidant, etc.
- The temperature stability of the prepared membranes was not examined.
- The mechanical properties were not examined
- The antifouling properties, flux recovey ratio could be further examined
- A major problem is that green tea is used for nutrition. Please, make an appropriate comment for the application of by products or wastes from green tea production.
- Characterize these membranes in terms i) of morphology (dense, microporous, etc) ii) physical state (glassy, semicrystalline, etc)
- Propose specific process for their application (microfiltration, osmosis, etc)In my opinion this work could be published after revision.
Round 2
Reviewer 1 Report
The authors made a good revision and answered all open question. The revised version now suitable for publication.
Reviewer 2 Report
Thank you for your revision